# Signal Analysis, Signal Demodulation and Numerical Simulation of a Quasi-Distributed Optical Fiber Sensor Based on FDM/WDM Techniques and Fabry-Pérot Interferometers

**DOI:** 10.3390/s19081759

**Published:** 2019-04-12

**Authors:** José Trinidad Guillen Bonilla, Héctor Guillen Bonilla, Verónica María Rodríguez Betancourtt, Antonio Casillas Zamora, María Eugenia Sánchez Morales, Lorenzo Gildo Ortiz, Alex Guillen Bonilla

**Affiliations:** 1Departamento de Electrónica, Centro Universitario de Ciencias Exactas e Ingenierías (C.U.C.E.I.), Universidad de Guadalajara, Blvd. M. García Barragán 1421, C. P. 44430, Guadalajara, Jalisco 44430, Mexico; 2Departamento de Matemáticas, Centro Universitario de Ciencias Exactas e Ingenierías (C.U.C.E.I.), Universidad de Guadalajara, Blvd. M. García Barragán 1421, C.P. 44430, Guadalajara, Jalisco 44430, Mexico; 3Departamento de Ingeniería de Proyectos, Centro Universitario de Ciencias Exactas e Ingenierías (C.U.C.E.I.), Universidad de Guadalajara, Blvd. M. García Barragán 1421, C. P. 44430, Guadalajara, Jalisco 44430, Mexico; hectorguillenbonilla@gmail.com (H.G.B.); antonio.czamora@academicos.udg.mx (A.C.Z.); 4Departamento de Química, Centro Universitario de Ciencias Exactas e Ingenierías (C.U.C.E.I.), Universidad de Guadalajara, Blvd. M. García Barragán 1421, C. P. 44430, Guadalajara, Jalisco 44430, Mexico; veronica.rbetancourtt@academicos.udg.mx; 5Departamento de Ciencias Tecnológicas, Centro Universitario de la Ciénega, Universidad de Guadalajara, Av. Universidad No. 1115, Lindavista, C. P. 47810, Ocotlán, Jalisco 47810, Mexico; eugenia.sanchez@cuci.udg.mx; 6Departamento de Física, Centro Universitario de Ciencias Exactas e Ingenierías (C.U.C.E.I.), Universidad de Guadalajara, Blvd. M. García Barragán 1421, C.P. 44430, Guadalajara, Jalisco 44430, Mexico; lorenzo.gildo@gmail.com; 7Departamento de Ciencias Computacionales, Centro Universitario de los Valles (CUValles), Universidad de Guadalajara, Ameca Km. 45.5, C.P., Ameca, Jalisco 46600, Mexico; alex.guillen@profesores.valles.udg.mx

**Keywords:** quasi-distributed optical fiber sensor, wavelength/frequency division multiplexing, Fabry-Pérot sensors, theoretical analysis, sensor simulation

## Abstract

In civil engineering quasi-distributed optical fiber sensors are used for reinforced concrete monitoring, precast concrete monitoring, temperature monitoring, strain monitoring and temperature/strain monitoring. These quasi-distributed sensors necessarily apply some multiplexing technique. However, on many occasions, two or more multiplexing techniques are combined to increase the number of local sensors and then the cost of each sensing point is reduced. In this work, a signal analysis and a new signal demodulation algorithm are reported for a quasi-distributed optic fiber sensor system based on Frequency Division Multiplexing/Wavelength Division Multiplexing (FDM/WDM) and low-precision Fabry-Pérot interferometers. The mathematical analysis and the new algorithm optimize its design, its implementation, improve its functionality and reduce the cost per sensing point. The analysis was corroborated by simulating a quasi-distributed sensor in operation. Theoretical analysis and numerical simulation are in concordance. The optimization considers multiplexing techniques, signal demodulation, physical parameters, system noise, instrumentation, and detection technique. Based on our analysis and previous results reported, the optical sensing system can have more than 4000 local sensors and it has practical applications in civil engineering.

## 1. Introduction

Many practical applications for fiber optic sensors require measuring the same parameter in different spatial locations. For such applications, quasi-distributed sensors can be applied. This sensor type has received significant research attention and it necessarily applies some multiplexing technique. Several basic multiplexing techniques such as time-division multiplexing, wavelength-, coherence-, frequency-, spatial-, and code-division multiplexing have been proposed, demonstrated and applied over the last 10 years [1,2,3,4,5,6,7,8,9,10]. However, the multiplexing capacities of many of these techniques are limited to only a few sensors due to various factors including cross-talk, limited power budget and wavelength bandwidth. To increase the multiplexing capacity, several hybrid systems using two or more basic techniques have been developed [11,12,13,14,15]. These hybrid systems offer us some benefits such as cost savings per sensing point and enhancement of the competitiveness of fiber-optic sensors in their rivalry with conventional sensor technologies.

In reference [12], a quasi-distributed sensor based on wavelength-frequency division multiplexing was experimentally proposed. The sensing system had two wavelength channels and each wavelength channel had four frequency channels. Thus, the quasi-distributed sensor had eight Fabry-Pérot interferometers where each interferometer acted as a local sensor. Later, the same sensing system was applied for temperature measurement. In this case, the optical sensing system had three wavelength channels and each channel had three frequency channels. There were then nine Fabry-Pérot fiber sensors along the single array of the optical fiber. The authors reported only a resolution of 0.125 °C and a sensitivity of $10\frac{pm}{{}^{\circ}C}$ [13]. However, each Fabry-Pérot sensor has its own resolution and its own sensitivity since each sensor has its own physical parameters as reported in references [16,17].

The optical system reported in [12,13], can be theoretically analyzed after stating three principal problems: (a) a sensing system based on frequency division multiplexing, (b) a sensing system based on wavelength division multiplexing and (c) a sensing system based on frequency-wavelength division multiplexing. Problems (a) and (b) were reported in [16,17]. Both theoretical analyses considered the capacity of the multiplexing technique, the instrumentation, the system noise, the signal demodulation, the detection technique and the local sensor properties. Both theoretical analyses had their design and functionality optimized, and both were corroborated by simulating the sensing system in operation. However, to our knowledge, there are no analytic analyses of the third problem (c), so the limitations of quasi-distributed sensors are not known. In this work, the third problem (c) is considered and therefore a quasi-distributed fiber sensor based on frequency-wavelength (FDM/WDM) division multiplexing is theoretically analyzed and numerically simulated. This analysis optimizes its design and its implementation. The analysis considers the capacity of multiplexing technique, local sensor properties, signal demodulation, detection technique, instrumentation and noise. To verify the theoretical analysis and a new demodulation algorithm, we performed a numerical simulation of a quasi-distributed sensor which has four wavelength channels and each channel has three frequency channels. Thus, the simulated quasi-distributed sensor has twelve Fabry-Pérot sensors. The local sensors are low-precision Fabry-Pérot interferometers which are formed by two identical Bragg gratings. All Bragg gratings must have low reflectivity (~1%) and the same length. From the numerical results, each Fabry-Pérot sensor has two resolutions due to the Fourier Domain Phase Analysis algorithm [18,19,20] as reported in [16,17], confirming the optimal functionality of our optical sensing system under study. Theory and simulation results are in concordance.

## 2. Optical System and its Reflection Spectrum

Figure 1 shows the optical system under study. The system consists of a light source, a 50/50 optical circulator, an optical spectrum analyzer (OSA spectrometer), a personal computer used for signal processing and the quasi-distributed sensor. Particularly, the quasi-distributed sensor consists of a serial arrangement of local sensors where each sensor is an interferometer formed by two identical Bragg gratings imprinted in a single-mode fiber. All Bragg gratings have low reflectivity, ≈1% and their lengths are similar. Each interferometric sensor acts as a low-precision Fabry-Perot interferometer [13,16,17]. In the quasi-distributed sensor, Wavelength Division Multiplexing (WDM) and Frequency Division Multiplexing (FDM) techniques were combined. The spatial resolution $L_{SR}$ eliminates ghost interferometers.

If the quasi-distributed fiber optic sensor does not have external perturbation due to temperature or strain, the reflection spectrum detected by the OSA spectrometer can be expressed as:(1)$$R_{T}\left(\lambda \right)=\begin{array}{c} R_{11}\left(\lambda \right)+R_{12}\left(\lambda \right)+\ldots +R_{1M}\left(\lambda \right)+\\ R_{21}\left(\lambda \right)+R_{22}\left(\lambda \right)+\ldots +R_{2M}\left(\lambda \right)+\\ \begin{array}{c} R_{31}\left(\lambda \right)+R_{32}\left(\lambda \right)+\ldots +R_{3M}\left(\lambda \right)+\\ \vdots \\ R_{K1}\left(\lambda \right)+R_{K2}\left(\lambda \right)+\ldots +R_{KM}\left(\lambda \right) \end{array} \end{array}$$
or:(2)$$R_{T}\left(\lambda \right)=\overset{K}{\underset{k=1}{\sum }}\overset{M}{\underset{m=1}{\sum }}R_{km}\left(\lambda \right)$$
where $R_{T}\left(\lambda \right)$ is the total reflection spectrum, $R_{km}\left(\lambda \right)\left(k=1,2,\ldots ,K\;y\;m=1,2,\ldots ,M\right)$ are the interference patterns produced by the Fabry-Pérot sensors. The light spectrum $R_{T}\left(\lambda \right)$ consists of a series of wavelength channels and each wavelength channel contains a series of frequency channels. Therefore, based on our previous work [16,17], the maximum number of interference patterns is:(3)$$K\times M=\frac{\lambda_{w}}{\Delta \lambda_{op}}\times \left(\frac{\lambda_{BGk}^{2}}{8nL_{BG}\Delta \lambda }\right)=\frac{\lambda_{max}-\lambda_{min}}{\Delta \lambda_{op}}\times \left(\frac{\lambda_{BGk}^{2}}{8nL_{BG}\Delta \lambda }\right)$$

K×M indicates the maximum number of local optical sensors and the parameters are: $\Delta \lambda_{op}$ is the dynamic range for all Fabry-Pérot sensors, $\lambda_{BGk}$ is the *k-th* Bragg wavelength (wavelength channel), n is the effective refraction index of the core, $L_{BG}$ is the length of the gratings, Δλ is the spectrometer resolution and $\lambda_{w}=\lambda_{max}-\lambda_{min}$ is the working interval: $\lambda_{max}$ is the maximum wavelength and $\lambda_{min}$ is the minimum wavelength. In this sensing system, all interference patterns have the same form:(4)$$R_{km}\left(\lambda \right)=2a_{km}\left[{\left(\frac{\pi n_{1}L_{BG}}{\lambda_{BGk}}\right)}^{2}sinc^{2}\left(\frac{2n_{1}L_{BG}\left(\lambda -\lambda_{BGk}\right)}{\lambda_{BGk}^{2}}\right)\right]\times \left[1+cos\left(\frac{4\pi nL_{FPkm}\left(\lambda -\lambda_{BGk}\right)}{\lambda_{BGk}^{2}}\right)\right]$$

The physical parameters are: $a_{nm}$, the amplitude factors, π the constant 3.1415, $n_{1}$ the amplitude of the effective refractive index modulation of the gratings, λ is the wavelength and $L_{FPkm}$ is *km–th* cavity length. Analysis of the interference pattern (4); each interference pattern consists of two functions: enveloping and modulating. The enveloped is a sinc function which is the reflection spectrum of the gratings, the width $\Delta_{BGk}$ is the spectral distance between its +1 and −1 zeros:(5)$$\Delta_{BGk}=\frac{\lambda_{BGk}^{2}}{n_{1}L_{BG}}$$

The modulating term of the cosine function whose frequency $\nu_{FPkm}$ is:(6)$$\nu_{FPkm}=\frac{2nL_{FPkm}}{\lambda_{BGk}^{2}}$$

The cosine function was produced due to the interference between two beams generated from the interferometry sensor. On the other hand, if the quasi-distributed sensor experiences an external perturbation due to temperature or strain, the physical variable affects the Fabry-Pérot sensor [18]. In contrast, the interference patterns have a small shift which is proportional to the magnitude of the physical parameter. In this case, the optical spectrum detected by the OSA spectrometer can be expressed as: (7)$$R_{T}\left(\lambda ,\delta \lambda \right)=\begin{array}{c} R_{11}\left(\lambda -\delta \lambda_{11}\right)+R_{12}\left(\lambda -\delta \lambda_{12}\right)+\ldots +R_{1M}\left(\lambda -\delta \lambda_{1M}\right)+\\ R_{21}\left(\lambda -\delta \lambda_{21}\right)+R_{22}\left(\lambda -\delta \lambda_{22}\right)+\ldots +R_{2M}\left(\lambda -\delta \lambda_{2M}\right)+\\ R_{31}\left(\lambda -\delta \lambda_{31}\right)+R_{32}\left(\lambda -\delta \lambda_{32}\right)+\ldots +R_{3M}\left(\lambda -\delta \lambda_{3M}\right)+\\ \vdots \\ R_{N1}\left(\lambda -\delta \lambda_{N1}\right)+R_{N2}\left(\lambda -\delta \lambda_{N2}\right)+\ldots +R_{NM}\left(\lambda -\delta \lambda_{KM}\right) \end{array}$$
or:(8)$$R_{T}\left(\lambda ,\delta \lambda \right)=\overset{K}{\underset{k=1}{\sum }}\overset{M}{\underset{m=1}{\sum }}R_{nm}\left(\lambda -\delta \lambda_{km}\right)$$
$R_{T}\left(\lambda ,\delta \lambda \right)$ is the reflection spectrum due to the external disturbances and $\delta \lambda_{km}$ is the *km*-th Bragg wavelength shift due to the measured change.

## 3. Frequency Spectrums

To calculate all frequency components of our optical spectrum $R_{T}\left(\lambda \right)$, we apply the Fourier transform:(9)$$R_{T}\left(\nu \right)=\overset{\infty }{\underset{-\infty }{\int }}R_{T}\left(\lambda \right)e^{-i2\pi \nu \lambda }d\lambda$$
$R_{T}\left(\nu \right)$ is the frequency spectrum for the optical signal detected by OSA spectrometer if and only if the quasi-distributed fiber optic sensor does not have external perturbations. Substituting Equation (2) and Equation (4) into Equation (9), the frequency spectrum can be calculated through:(10)$$\begin{array}{c} R_{T}\left(\nu \right)=\overset{\infty }{\underset{-\infty }{\int }}\overset{K}{\underset{k=1}{\sum }}\overset{M}{\underset{m=1}{\sum }}2a_{km}\left[{\left(\frac{\pi n_{1}L_{BG}}{\lambda_{BGk}}\right)}^{2}sinc^{2}\left(\frac{2n_{1}L_{BG}\left(\lambda -\lambda_{BGk}\right)}{\lambda_{BGk}^{2}}\right)\right]\times \\ \left[1+cos\left(\frac{4\pi nL_{FPkm}\left(\lambda -\lambda_{BGk}\right)}{\lambda_{BGk}^{2}}\right)\right]e^{-i2\pi \nu \lambda }d\lambda \end{array}$$
invoking the Fourier transform properties, convolution properties, series properties and using the identities: $cos^{2}\left(\varphi \right)=\frac{1}{2}\left[1+cos\left(2\varphi \right)\right]$ and $cos\left(\varphi \right)=\frac{e^{i\varphi }+e^{-i\varphi }}{2}$, then solving the frequency spectrum $R_{T}\left(\nu \right)$ is:(11)$$R_{T}\left(\nu \right)=\overset{K}{\underset{k=-K}{\sum }}\overset{M}{\underset{m=-M}{\sum }}R_{km}\left(\nu \right)=\overset{K}{\underset{k=-K}{\sum }}\overset{M}{\underset{m=-M}{\sum }}c_{km}tri\left(\frac{\nu -\nu_{FPkm}}{\nu_{BGk}}\right)$$
$R_{T}\left(\nu \right)$ is two series of triangle functions and the function tri(x) has the following definition $tri\left(x\right)=\{\begin{array}{cc} 1-\left|x\right| & \left|x\right|\leq 1\\ 0 & otherwise \end{array}$, $c_{km}$ are amplitude factors, $\nu_{FPkm}$ is the *km*-th central frequency which was defined by Equation (6) and $\nu_{BGk}$ is the bandwidth:(12)$$\nu_{BGk}=\frac{4n_{1}L_{BG}}{\lambda_{BGk}^{2}}$$

From Equation (11), the spectrum $R_{T}\left(\nu \right)$ consists of 2(K×M)+1 triangle functions (frequency components) and their central frequencies are between $-\nu_{FPkm}$ and $\nu_{FPkm}$. The component $\nu_{FP00}=0$ contains information from all Fabry-Perot sensors while the lateral components contain information from each Fabry-Perot sensor. Positive semi-plane and negative semi-plane contain the same information. If the quasi-distributed sensor has external perturbation, the perturbed frequency spectrum can be calculated through:(13)$$R_{T}\left(\nu ,\delta \lambda \right)=\overset{\infty }{\underset{-\infty }{\int }}R_{T}\left(\lambda ,\delta \lambda \right)e^{-i2\pi \nu \lambda }d\lambda$$

Substituting Equation (8) into Equation (13):(14)$$R_{T}\left(\nu ,\delta \lambda \right)=\overset{\infty }{\underset{-\infty }{\int }}\overset{K}{\underset{k=1}{\sum }}\overset{M}{\underset{m=1}{\sum }}R_{nm}\left(\lambda -\delta \lambda_{km}\right)e^{-i2\pi \nu \lambda }d\lambda$$
invoking the shift property $\int_{-\infty }^{\infty }R\left(\lambda -\delta \lambda \right)e^{-i2\pi \nu \lambda }d\lambda =R\left(\lambda \right)e^{-i2\pi \nu \delta \lambda }$ and solving the transformation, the perturbed frequency spectrum will be:(15)$$R_{T}\left(\nu ,\delta \lambda \right)=\overset{K}{\underset{k=-K}{\sum }}\overset{M}{\underset{m=-M}{\sum }}R_{km}\left(\nu \right)e^{-i2\pi \nu \delta \lambda_{km}}$$

Analyzing the Equation (15), the spectrum $R_{T}\left(\nu ,\delta \lambda \right)$ is the product of $R_{T}\left(\nu \right)$ and two sets of phases. We notice that the phases contain the information about the perturbations and then the Fourier Domain Phase Analysis (FDPA) algorithm can be applied for the signal demodulation [18,19,20].

## 4. Sensor Conditions

To obtain the optimal signal demodulation is important to know the sensor limits. Therefore, we mention some requirements for the optical sensing system presented in Figure 1 [16,17,18]:

*First condition*: The cavity length $L_{FP}$ should be in the interval of:(16)$$2L_{BG}\leq L_{FP}\leq \frac{\lambda_{BGk}^{2}}{4n\Delta \lambda }$$

This condition ensures that the OSA spectrometer is able to detect the optical signal $R_{T}\left(\lambda \right)$ and there is not overlapping between the components $\nu_{FP11}$ and $\nu_{FP00}$.

*Second condition*: The Bragg gratings should have approximately the same length and their reflectivity is ≈1%. Thus, the cross-talk noise is eliminated.

*Third Condition*: The spatial resolution should be given by
(17)$$L_{FP}>\frac{\lambda_{BGk}^{2}}{4n\Delta \lambda }$$

This condition eliminates ghost interferometers.

*Fourth condition*: The minimum distance between centers of gratings for the shortest interferometers is given by:(18)$$L_{BGd}=2L_{BG}$$

This condition eliminates overlapping between triangle functions.

*Fifth condition*: The number of samples is given by:(19)$$N=\frac{\lambda_{w}}{\delta \lambda }=\frac{4n\lambda_{w}\left(2L_{BG}+L_{FPKM}\right)}{\lambda_{BG1}^{2}}$$
where $\lambda_{BG1}$ is the first Bragg wavelength (first wavelength channel) and $L_{FPKM}$ is the maximum cavity length (last frequency channel and last wavelength channel). The relation (19) satisfices the Nyquist theorem, considers the detection technique, the instrumentation and the sensor parameters.

*Sixth condition*: The maximum number of Fabry-Pérot sensors is given by Equation (3). This Equation considers both multiplexing techniques (WDM and FDM), sensor parameters, optical source, optical instrumentation and signal demodulation. If the six conditions are satisfied, then the sensing system would have the optimal operation.

## 5. Signal Demodulation

Figure 2 shows, schematically, the signal demodulation procedure for the sensing system presented in Figure 1. The signal demodulation applies the Fourier Domain Phases Analysis (FDPA) algorithm and two banks of filters. The FDPA algorithm was described and also was applied in references [16,17,18,19,20]. The first bank of K filters is defined as [16]:(20)$$F\left(\lambda \right)=\overset{K}{\underset{k=1}{\sum }}rect\left(\frac{\lambda -\lambda_{BGk}}{\Delta \lambda_{op}}\right)$$

The bank is a series of rect(·) functions in the wavelength domain where the rect function is defined as $rect\left(\lambda \right)=\{\begin{array}{cc} 1 & \left|\lambda \right|\leq \frac{\Delta \lambda_{op}}{2}\\ 0 & \left|\lambda \right|>\frac{\Delta \lambda_{op}}{2} \end{array}$, $\Delta \lambda_{op}$ is the dynamic range and $\lambda_{BGk}$ is the wavelength channel. The second bank of M filters is given by [17]:(21)$$F\left(\nu \right)=\overset{M}{\underset{m=1}{\sum }}rect\left(\frac{\nu -\nu_{FPkm}}{\nu_{BGk}}\right)$$

The bank is a series of rect(·) functions in frequency domain and each rect function definition is by $rect\left(\nu \right)=\{\begin{array}{cc} 1 & \left|\nu \right|\leq \frac{\nu_{BGk}}{2}\\ 0 & \left|\nu \right|>\frac{\nu_{BGk}}{2} \end{array},$
$\nu_{BGk}$ is the bandwidth and $\nu_{FPkm}$ is the centering frequency.

From Figure 2, the signal demodulation algorithm consists of two phases: calibration and measurement. In the calibration phase, the references are estimated and five steps are required: (a) the optical signal $R_{T}\left(\lambda \right)$ is acquired with the OSA spectrometer; (b) using the bank of K filters is filtered a wavelength channel, $R_{m}\left(\lambda \right)=F\left(\lambda \right)R_{T}\left(\lambda \right)$; (c) the frequency spectrum $R_{m}\left(\nu \right)$ is estimated through: (22)$$R_{m}\left(\nu \right)=\overset{\infty }{\underset{-\infty }{\int }}R_{m}\left(\lambda \right)e^{-i2\pi \nu \lambda }d\lambda$$

$R_{m}\left(\nu \right)$ is a series of triangle functions [13,16]; (d) a frequency channel (a triangle function) ${\tilde{R}}_{m}\left(\nu \right)$ is filtered using ${\tilde{R}}_{m}\left(\nu \right)=F\left(\nu \right)R_{m}\left(\nu \right)$ and finally (e) its complex conjugate is estimated ${\tilde{R}}_{m}^{*}\left(\nu \right)$ where the symbol * indicates the complex conjugate. ${\tilde{R}}_{m}^{*}\left(\nu \right)$ is the reference for one Fabry-Pérot sensor and then steps a-d are essential for each interferometer sensor.

In the measurement phase, eight steps are required: (a) the optical spectrum $R_{T}\left(\lambda ,\delta \lambda \right)$ is acquired using the OSA spectrometer; (b) applying the bank of K filters is filtered the signal $R_{m}\left(\lambda ,\delta \lambda \right)=F\left(\lambda \right)R_{T}\left(\lambda ,\delta \lambda \right)$; (c) the frequency spectrum $R_{m}\left(\nu ,\delta \lambda \right)$ is determined by:(23)$$R_{m}\left(\nu ,\delta \lambda \right)=\overset{\infty }{\underset{-\infty }{\int }}R_{m}\left(\lambda ,\delta \lambda \right)e^{-i2\pi \nu \lambda }d\lambda$$
$R_{m}\left(\nu ,\delta \lambda \right)$ is a series of triangle functions; (d) a frequency channel (a triangle function) ${\tilde{R}}_{m}\left(\nu ,\delta \lambda \right)$ is filtered developing the operation ${\tilde{R}}_{m}\left(\nu ,\delta \lambda \right)=F\left(\nu \right)R_{m}\left(\nu ,\delta \lambda \right)$; (e) the relative phase $\varphi_{m,rel}$ is calculated using both spectra ${\tilde{R}}_{m}^{*}\left(\nu \right)$ and ${\tilde{R}}_{m}\left(\nu ,\delta \lambda \right)$; (f) the ambiguity 2πP is eliminated through a linear regression and then the absolute phase $\varphi_{m,abs}$ is calculated; (g) an adaptive filter is applied, and (h) the Bragg wavelength shift $\delta \lambda_{m}$ is calculated. 

To minimize the noise influence and provide the best estimate, the absolute phase is multiplied by a set of coefficients (step g). Those coefficients act as adaptive filters [20].

## 6. Numerical Results and Discussion

### 6.1. Parameters

To verify our signal analysis and our signal demodulation algorithm, a quasi-distributed fiber optic sensor based on the wavelength/frequency multiplexing techniques and low-precision Fabry-Pérot interferometers was numerically simulated. The simulated quasi-distributed sensor can be seen in Figure 1. The sensing system consists of four wavelength channels and each wavelength channel has three frequency channels, therefore, the sensing system has twelve Fabry-Pérot sensors. Their physical parameters are listed in Table 1. 

Discrete spectra were simulated applying these parameters. Noise was simulated by adding pseudorandom numbers with Gaussian distribution to those samples, the interval was from $\sqrt{SNR}=10^{0}$ to $\sqrt{SNR}=10^{4}$. Typical Bragg gratings with rectangular profiles, a refractive index modulation were used. The number of samples was 4096. Given that, the Fast Fourier transform algorithm was considered. The spatial resolution was $L_{SR}=50\;\mathrm{cm}$. For each Fabry-Pérot sensor, the reference spectrum and 50 measurements were simulated. These measurements were within the interval in Table 2. In the simulation a GHIA computer with 8 GB RAM memory and CPU frequency 3.6 GHz was used.

### 6.2. Reflection Spectrum

Applying the parameters presented in Section 6.1, the optical signal $R_{T}\left(\lambda \right)$ was computed. The normalized spectra can be observed in Figure 3. The composite reflection spectrum of a multi-point Fabry-Perot sensor is a superposition of reflection spectra of all local sensors. From Figure 3, the first wavelength channel $\left(\lambda_{BG1}=1536\;\mathrm{nm}\right)$ has its bandwidth of $\Delta_{BG1}\approx 3.2\;\mathrm{nm}$, the second wavelength channel $\left(\lambda_{BG2}=1542\;\mathrm{nm}\right)$ has its bandwidth of $\Delta_{BG2}\approx 3.24\;\mathrm{nm}$, the third wavelength channel $\left(\lambda_{BG3}=1548\;\mathrm{nm}\right)$ has its bandwidth of $\Delta_{BG2}\approx 3.3\;\mathrm{nm}$ and the fourth wavelength channel $\left(\lambda_{BG4}=1554\;\mathrm{nm}\right)$ has its bandwidth of $\Delta_{BG4}\approx 3.308\;\mathrm{nm}$. Each wavelength channel has its own value and they have a very small variation between them. The working interval is $\lambda_{w}=24\;\mathrm{nm}$: $\lambda_{max}=1556\;\mathrm{nm}$ and $\lambda_{min}=1532\;\mathrm{nm}$.

### 6.3. Frequency Spectrum

Figure 4 shows the positive components for our frequency spectrum $R_{T}\left(\nu \right)$. Analyzing Figure 4, the quasi-distributed fiber optic sensor has twelve Fabry-Pérot sensors and each local sensor produces one triangle function (frequency component, peaks). The red color corresponds to the frequency channels, $\nu_{FP11}-\nu_{FP13}$. Three peaks have the same bandwidth $\nu_{BG1}=1.23$ and their central frequencies are $\nu_{FP11}=9.90$, $\nu_{FP12}=19.80$, and $\nu_{FP13}=39.60$. The blue color corresponds to the frequency channels $\nu_{FP21}-\nu_{FP23}$. Again, three peaks have the same bandwidth $\nu_{BG2}=1.22$ and their central frequencies are $\nu_{FP21}=6.14$, $\nu_{FP22}=14.73$ and $\nu_{FP23}=29.47$. The black color corresponds to the channels $\nu_{FP31}-\nu_{FP33}$. Now, the bandwidth is $\nu_{BG3}=1.218$ and their central frequencies are $\nu_{FP31}=8.52$, $\nu_{FP32}=20.71$ and $\nu_{FP33}=32.90$. Finally, the cyan color corresponds to the channels $\nu_{FP41}-\nu_{FP43}$. The bandwidth is $\nu_{BG4}=1.209$ and the central frequencies are $\nu_{FP41}=14.50$, $\nu_{FP42}=26.60$ and $\nu_{FP43}=45.94$, respectively. The bandwidths $\nu_{BG1}\ldots \nu_{BG4}$ have small variations because each wavelength channel has its own Bragg wavelength, see Equation (12). Each interference pattern has its own frequency because each Fabry-Pérot sensor has its own cavity length and wavelength channel (see Equation (6)), see Table 1 and Figure 4.

Here, each frequency channel contains information from a specific Fabry-Pérot sensor and then the demodulation algorithm described in Section 5 can be applied.

### 6.4. Numerical Results

Applying the demodulation algorithm described in Section 5 and using the parameters presented in Section 6.1, the quasi-distributed fiber optic sensor (Figure 1) was numerically simulated. Our numerical results are shown in Figure 5. We present the behavior of demodulation errors vs. signal-to-noise rate SNR12. If demodulation errors are called the “resolution” as in our previous work [16,17], then all low-precision Fabry-Pérot sensors have two resolutions. Both resolutions are possible because the signal demodulation (Figure 2) is based on the FDPA algorithm and this algorithm evaluates the Bragg wavelength shift twice. Observing Figure 5, each Fabry-Pérot sensor has its own high resolution; however each wavelength channel produces one low resolution because this resolution does not depend on the cavity length. Additionally, the transition from high resolution to low resolution was reported [16,17,19], again, our numerical results were presented. These results corroborate our signal analysis and our demodulation algorithm.

For example: if the OSA spectrometer has Δλ=10 pm (a typical value), the broadband source has $\lambda_{w}=100\;\mathrm{nm}$ and as $\Delta \lambda_{op}=\lambda_{BG4}-\lambda_{BG3}=1554\;\mathrm{nm}-1548\;\mathrm{nm}=6\;\mathrm{nm}$ (see Figure 3), the quasi-distributed sensor will have their limits as Table 3 illustrates.

From Table 1, Table 2 and Table 3, the simulated quasi-distributed sensor satisfies the instrumentation and the signal requirements. Observing Table 1 and Table 3 and Figure 3, Figure 4 and Figure 5, the numerical results are in concordance with the theory. Thus, we confirm our theoretical analysis and our new demodulation algorithm. The numerical results are shown in Figure 5. The theoretical analysis and our numerical results are in concordance with experimental results reported by Shlyagin et al. [12] and Della-Tamin et al. [13]. Then, WDM/FDM techniques can be implemented on low-precision Fabry-Pérot sensors and our new algorithm demodulates its optical signal. The presented study optimizes significantly the quasi-distributed sensor implementation, its design and the sensibility of all local sensors.

### 6.5. Discussion 

Based on our signal analysis and our numerical results, the quasi-distributed sensor would be built through WDM/FDM techniques and low-finesse Fabry-Pérot interferometers. The theoretical analysis optimizes the quasi-distributed sensor presented in Figure 1 which was experimentally proposed in [12,13]. The optimization considers the multiplexing technique, the optical instrumentation, the detection technique, the local sensor properties, the noise system, the spatial resolution and the signal demodulation. Observing Figure 5, the quasi-distributed sensor has good functionality because each Fabry-Pérot sensor is performing very well and our new algorithm can demodulate the optical signal. The Fabry-Pérot sensors have high resolution when the signal-to-noise rate (SNR) is big, but the same sensors have low resolution if and only if the signal-to-noise rate is low (the optical system has many noise). The threshold value between both resolutions (high and low resolutions) can be determined by:(24)$$\sigma_{envkm}=\frac{\lambda_{BGk}^{2}}{12nL_{FPkm}}$$
where $\sigma_{envkm}$ is the low resolution for the km–th Fabry-Pérot sensor. This value depends of the Bragg wavelength and the cavity length. Applying the parameters presented in Table 1, the threshold values were calculated as Table 4 shows.

From Table 4, each Fabry-Pérot sensor has its own threshold value. Combining the multiplexing techniques (WDM and FDM), the number of local sensors was increased for the optical sensing system presented in Figure 1. For example: if the frequency division multiplexing has forty frequency channels (M = 40) [16] and if the wavelength division multiplexing has 100 wavelength channels (K = 100) [21,22], then our sensing system would have MXK = 40 × 100 = 4000 local sensors. This confirms that the quasi-distributed sensor has high capacity to measure some physical parameters along the optic fiber. Additionally, applying the conditions 1–6 described in Section 4, ghost interferometers, cross-talk noise and overlapping between two frequency components are eliminated, proving that the sensing system will have optimal functionality.

Figure 1 showed a serial array of interferometer sensors and the topology permits the measurements along of a cable. For the measurement over any surface, a serial/parallel topology can be applied. In this case, the optical system consists of an optical broadband source, an OSA spectrometer, a 2XK-splitter and a quasi-distributed sensor. The sensor would have K-fiber optics and each fiber optic can have M-Fabry-Perot interferometers. The proposed optical sensor can be observed in Figure 6. Their practical application can be on: civil engineering, mechanical engineering, military applications, civil protection and disaster risk reduction.

The quasi-distributed fiber sensor presented in this work applied the direct spectroscopic detection (DSD) technique. On the one hand, this detection method has a simple configuration and it just basically required an optical broadband source, an optical circulator, an optical spectrometer analyzer (OSA spectrometer) and a PC computer, see Figure 1. On the other hand, the authors described that the Optical Frequency Domain Reflectometry (OFDR) fell into two main categories [23]: incoherent (or direct detection) OFDR (I-OFDR) and the coherent OFDR (C-OFDR). From the first category, there are four derivations known as Network Analysis OFDR (NA-OFDR) [24], Incoherent Frequency-Modulate Continuous Wave (I-FMCW) [25], Step-frequency method [26] and Sweep Frequency method [27]. Each method was previously applied on other sensing systems. Comparing the sensing system presented in Figure 1 with the sensing systems reported by the authors of reference [23], the direct spectroscopic detection technique has some benefits, for example: simple configuration, low cost, less complexity and fewer instruments used, therefore, our sensing system has some advantages over other detection techniques.

Our future work is in the following direction: the transition between both resolutions could be determined using the physical parameters, instrumentation, the FDPA algorithm and noise. The quasi-distributed sensor application would be in this other direction.

## 7. Conclusions

In this work, a quasi-distributed fiber sensor was theoretically analyzed and also was numerically simulated. The quasi-distributed sensor was based on wavelength/frequency division multiplexing and low-finesse Fabry-Pérot interferometers. Theory and simulation were in concordance. During the analysis and the simulation, we considered the signal processing, multiplexing techniques, the optical instrumentation, the system noise, the detection technique and the local sensor parameters. The signal analysis and signal demodulation algorithm optimize the sensor implementation while the numerical simulation demonstrated its excellent functionality. From our numerical results, we confirmed that each Fabry-Pérot sensor has two resolutions since the Fourier Domain Phase Analysis algorithm makes two evaluations of the Bragg wavelength shift as previously reported by us. This quasi-distributed sensor finds many potential industrial applications due to its functionality, low cost by sensing point, high resolution and high sensitivity.

## Figures and Tables

**Figure 1 sensors-19-01759-f001:**
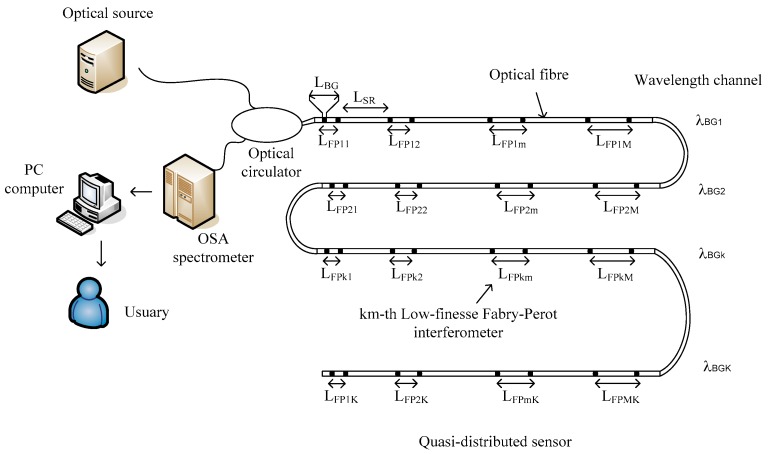
The quasi-distributed sensor based on Frequency Division Multiplexing/Wavelength Division Multiplexing (FDM/WDM) techniques and low-finesse Fabry-Perot interferometers.

**Figure 2 sensors-19-01759-f002:**
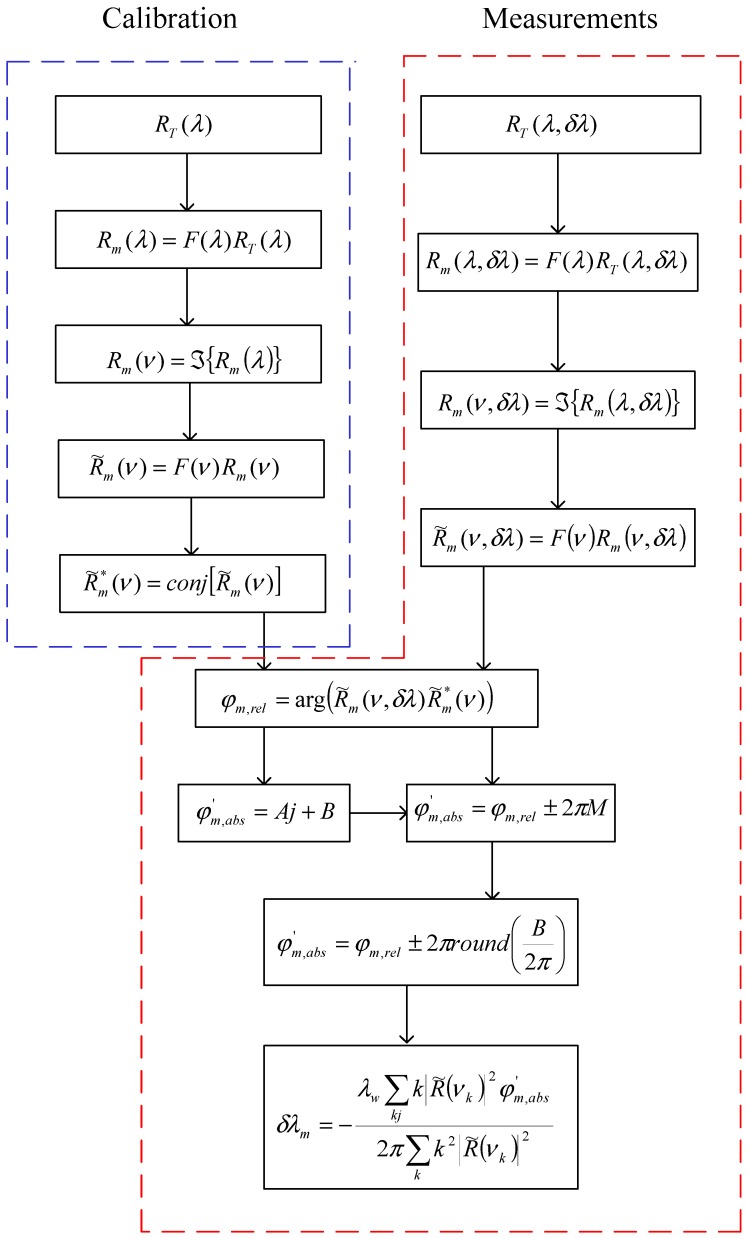
Signal demodulation procedure represented schematically where the symbol ℑ indicates the Fourier transform, *conj* indicates the complex conjugate and *k* is the samples.

**Figure 3 sensors-19-01759-f003:**
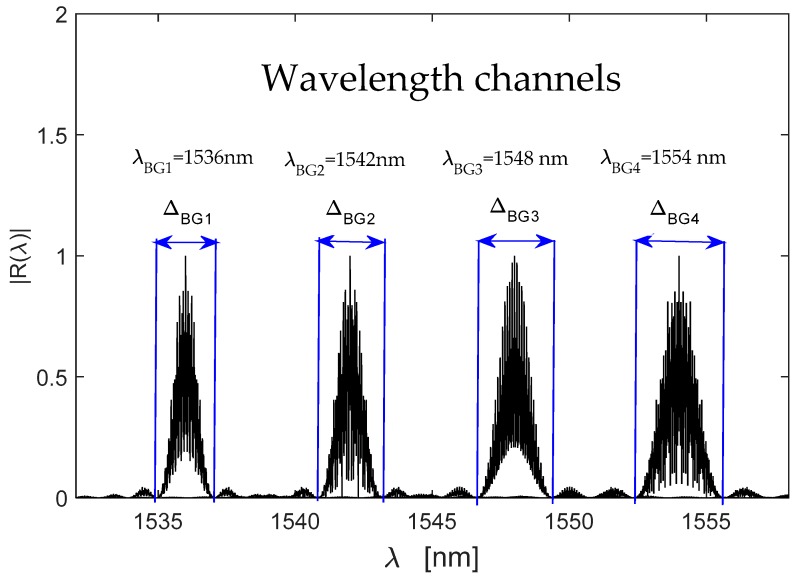
Simulated optical spectrum $\left|R_{T}\left(\lambda \right)\right|$ obtained for the quasi-distributed sensor: the symbol |·| indicates the normalization.

**Figure 4 sensors-19-01759-f004:**
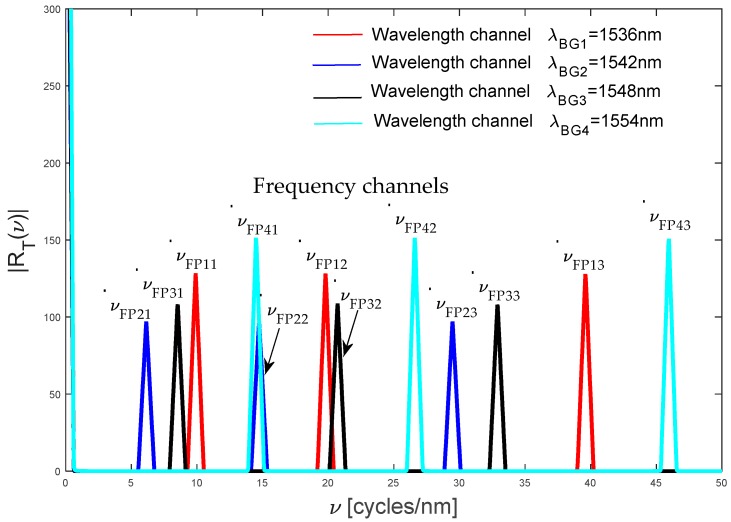
Frequency channels generated by the quasi-distributed sensor.

**Figure 5 sensors-19-01759-f005:**
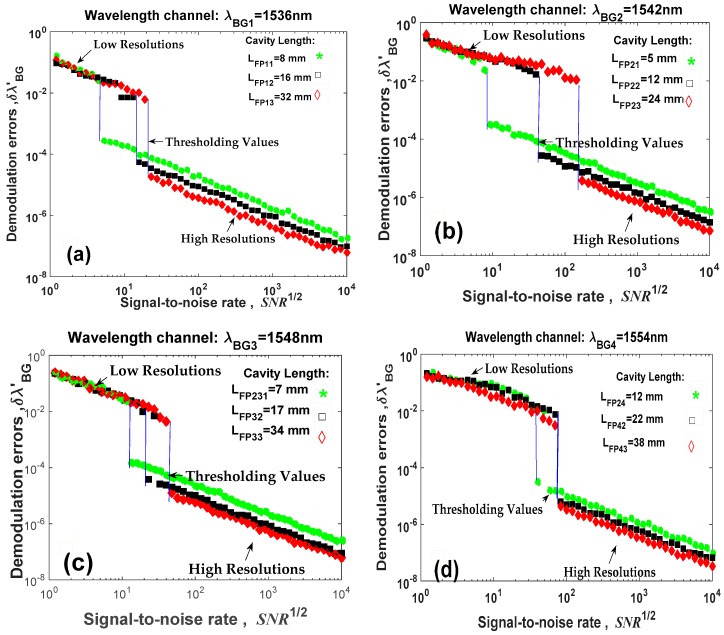
Numerical results obtained for the quasi-distributed fiber optic sensor.: (**a**) Wavelength channel $\lambda_{BG1}=1536\;nm$ and frequency channels $\nu_{FP11}=9.90$, $\nu_{FP12}=19.80$, and $\nu_{FP13}=39.60$; (**b**) Wavelength channel $\lambda_{BG2}=1542\;nm$ and frequency channels $\nu_{FP21}=6.14$, $\nu_{FP22}=14.73$ and $\nu_{FP23}=29.47$; (**c**) Wavelength channel $\lambda_{BG3}=1548\;nm$ and frequency channels $\nu_{FP31}=8.52$, $\nu_{FP32}=20.71$ and $\nu_{FP33}=32.90$; (**d**) Wavelength channel $\lambda_{BG3}=1554\;nm$ and frequency channels $\nu_{FP41}=14.50$, $\nu_{FP42}=26.60$ and $\nu_{FP43}=45.94$.

**Figure 6 sensors-19-01759-f006:**
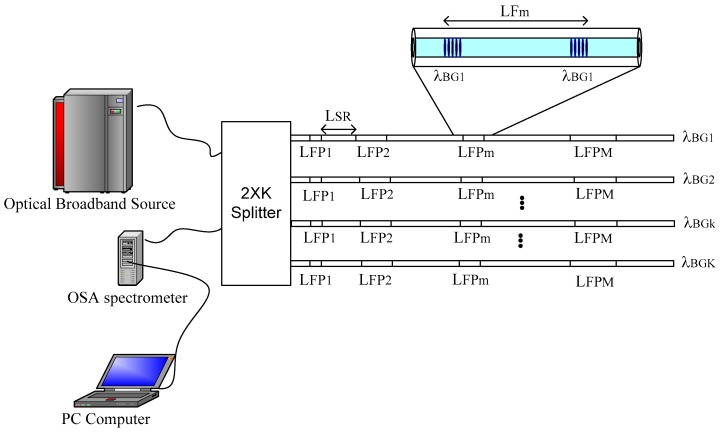
Quasi-distributed sensor based on the parallel-serial topologies.

**Table 1 sensors-19-01759-t001:** Simulated quasi-distributed optic fiber sensor parameters.

Wavelength Channel $\lambda_{BGk}$		Frequency Channel $\nu_{FPkm}$		Fabry-Pérot Sensors Parameters
Channel k	Value [nm]	Channel m	Value [Cycles/nm]	
1	1536	1	$\nu_{FP11}=9.90$	*n* = 1.46
				*L_FP11_* = 8 mm
				*L_BG_* = 0.5 mm
		2	$\nu_{FP12}=19.80$	*n* = 1.46
				*L_FP12_* = 16 mm
				*L_BG_* = 0.5 mm
		3	$\nu_{FP13}=39.60$	*n* = 1.46
				*L_FP13_* = 32 mm
				*L_BG_* = 0.5 mm
2	1542	1	$\nu_{FP21}=6.14$	*n* = 1.46
				*L_FP21_* = 5 mm
				*L_BG_* = 0.5 mm
		2	$\nu_{FP22}=14.73$	*n* = 1.46
				*L_FP22_* = 12 mm
				*L_BG_* = 0.5 mm
		3	$\nu_{FP23}=29.47$	*n* = 1.46
				*L_FP23_* = 24 mm
				*L_BG_* = 0.5 mm
3	1548	1	$\nu_{FP31}=8.52$	*n* = 1.46
				*L_FP31_* = 7mm
				*L_BG_* = 0.5mm
		2	$\nu_{FP32}=20.71$	*n* = 1.46
				*L_FP32_* = 17 mm
				*L_BG_* = 0.5 mm
		3	$\nu_{FP33}=32.90$	*n* = 1.46
				*L_FP33_* = 27 mm
				*L_BG_*= =.5 mm
4	1554	1	$\nu_{FP41}=14.50$	*n* = 1.46
				*L_FP41_* = 12 mm
				*L_BG_* = 0.5 mm
		2	$\nu_{FP42}=26.60$	*n* = 1.46
				*L_FP42_* = 22mm
				*L_BG_* = 0.5 mm
		3	$\nu_{FP43}=45.94$	*n* = 1.46
				*L_FP43_* = 38 mm
				*L_BG_* = 0.5 mm

**Table 2 sensors-19-01759-t002:** Applied displacement to each Fabry-Pérot sensor in the numerical simulation.

Wavelength Channel $\lambda_{BGk}$		Frequency Channel $\nu_{FPkm}$	Displacement Applied to Each Fabry-Pérot Sensor, $\delta \lambda_{km}$
Fabry-Pérot Sensor $S_{km}$	Value [nm]	Channel m (Value)	
$S_{11}$	1536	1 (9.90)	0–0.2
$S_{12}$		2 (19.80)	0–0.4
$S_{13}$		3 (39.60)	0–0.8
$S_{21}$	1542	1 (6.14)	0–0.32
$S_{22}$		2 (14.73)	0–0.24
$S_{23}$		3 (29.47)	0–0.85
$S_{31}$	1548	1 (8.52)	0–0.57
$S_{32}$		2 (20.71)	0–0.12
$S_{33}$		3 (32.90)	0–0.28
$S_{41}$	1554	1 (14.50)	0–0.7
$S_{42}$		2 (26.60)	0–0.23
$S_{43}$		3 (45.94)	0–0.77

Note: $S_{km}$ indicates the *km*–th Fabry-Perot sensor of the quasi-distributed sensor.

**Table 3 sensors-19-01759-t003:** Quasi-distributed sensor limits $\left(\Delta \lambda =10{\;\mathrm{pm}\;\mathrm{and}\;\lambda }_{w}=100\;\mathrm{nm}\right)$.

Parameter	Value	Equation
K	16 [wavelength channels]	Equation (3)
M	40 [Frequency channels]	Equation (3)
K×M	640 [Fabry-Pérot sensors]	Equation (3)
$2L_{BG}\leq L_{FP}\leq \frac{\lambda_{BG4}^{2}}{4n\Delta \lambda }$	$1\leq L_{FP}\leq \frac{\lambda_{BG4}^{2}}{4n\Delta \lambda }\;\left[\mathrm{mm}\right]$	Equation (16)
$L_{FP}>\frac{\lambda_{BG4}^{2}}{4n\Delta \lambda }$	$L_{FP}>\frac{\lambda_{BG4}^{2}}{4n\Delta \lambda }\;\left[\mathrm{mm}\right]$	Equation (17)
$L_{BGd}=2L_{BG}$	1 [mm]	Equation (18)

**Table 4 sensors-19-01759-t004:** Threshold value calculated for each Fabry-Pérot sensor.

Local Sensors, $S_{km}$	Parameter $\sigma_{envkm}$	Local Sensors, $S_{km}$	Parameter $\sigma_{envkm}$
$S_{11}$	$\sigma_{env11}=0.016$	$S_{31}$	$\sigma_{env31}=0.019$
$S_{12}$	$\sigma_{env12}=0.0084$	$S_{32}$	$\sigma_{env32}=0.0079$
$S_{13}$	$\sigma_{env13}=0.0042$	$S_{33}$	$\sigma_{env33}=0.005$
$S_{21}$	$\sigma_{env21}=0.027$	$S_{41}$	$\sigma_{env41}=0.011$
$S_{22}$	$\sigma_{env22}=0.011$	$S_{42}$	$\sigma_{env42}=0.006$
$S_{23}$	$\sigma_{env23}=0.0056$	$S_{43}$	$\sigma_{env43}=0.003$

$S_{km}$*km*-th Fabry-Pérot sensor (*k* = 1, 2, 3, 4 and *m* = 1, 2, 3).

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
