# Peer review of "Signal Analysis, Signal Demodulation and Numerical Simulation of a Quasi-Distributed Optical Fiber Sensor Based on FDM/WDM Techniques and Fabry-Pérot Interferometers"

_sensors, 2019, doi:10.3390/s19081759_

Reviewer 1 Report

Results may be of interests to readers. However, authors should address the following  aspects:

beside lower cost per sensing point, are there additional benefits the demonstrated quasi-distributed sensing system offers, particularly compared to other well studied and applied fiber based sensing technology such as OFDR ?

Line 103, explain "no external perturbation" conditions or when the expression (1) is a valid approximation

Figure 6, what type light source is used in the demonstrated  sensing network.

Author Response

Response to Reviewer 1 comments:

Results may be of interests to readers. However, authors should address the following aspects:

1.      beside lower cost per sensing point, are there additional benefits the demonstrated quasi-distributed sensing system offers, particularly compared to other well studied and applied fiber based sensing technology such as OFDR?

Answer:

The reviewer does an interest question.

The sensing system presented in this work does have additional benefits related to some OFDR techniques, for example: the instrumentation required, configuration system, complexity and cost.

Correction:

In the article, lines 335-350 (page 14) were written and Figure 7 was collocated. This new paragraph is supported by the new references 23-27.

2.      Line 103, explain "no external perturbation" conditions or when the expression (1) is a valid approximation

Answer:

 The expression (1) is a valid approximation when the quasi-distributed sensor does not have external perturbation due to temperature, strain or other physical variables. If the quasi-distributed sensor has an external perturbation, then its reflection spectrum will be the expression (7).

Correction:

The sentence between lines 103-104 (paragraph 2, page 3),

“If the quasi-distributed fiber optic sensor does not have external perturbation, the reflection spectrum detected by the OSA spectrometer can express”

was change by

If the quasi-distributed fiber optic sensor does not have external perturbation due to the temperature or strain, the reflection spectrum detected by the OSA spectrometer can express

3.      Figure 6, what type light source is used in the demonstrated sensing network.

Answer:

An optical broadband source can be applied in optical sensing system of Figure 6. This optical source is required because each wavelength channel has its our bandwidth and all wavelength channels must be within of the working interval                                               , see Figure 6.

Correction:

In Figure 6, Optical Source was changed by Optical Broadband Source

In line 330, optical source was changed by optical broadband source

Reviewer 2 Report

The paper reports a method for interrogating quasi-distributed Fabry-Perot interferometers fabricated with pairs of FBGs by using a demultiplexing technique comprising the combination of WDM and FDM schemes. The spectral shifts induced by the applied stimuli are detected by means of Fourier-domain phase analysis, making it possible to assess the response of K wavelength channels and M frequency channels, yielding an array of K´M sensing units.

The topic addressed in this manuscript in very interesting as it proposes an alternative for expanding the capability of quasi-distributed fiber sensor networks. Moreover, the detections limits for this technique are discussed in terms of the experimental apparatuses characteristics, which is useful for the design of experiments.

Although the paper has its merits, in my opinion, the manuscript is not acceptable for publication in the present form for the following reasons:

The English must be extensively revised. For      example, in some cases both American and British English spelling appears      in the text (‘fiber’ and ‘fibre’), so the authors should keep it uniform.      Another example, in Equation (11), the definition of tri function, the      expression “otro caso” must be translated into English;

Although the theory and the simulation      results are compatible, the authors must present experimental validation      of the interrogation method in order to demonstrate its feasibility. There      are several issues that must be considered in practical measurements, such      as the algorithm processing time, the robustness to environmental noise,      the imperfections in the FBG and FPI dimensions, etc.

Therefore, I strongly advise the authors to perform the experiments and submit the paper again in order to make it suitable for a broader audience.

Author Response

Response to Reviewer 2 comments:

1        The English must be extensively revised. For example, in some cases both American and British English spelling appears in the text (‘fiber’ and ‘fibre’), so the authors should keep it uniform. Another example, in Equation (11), the definition of tri function, the expression “otro caso” must be translated into English;

Answer (and correction):

We are very grateful to the reviewer for his wise suggestions and for letting us see our mistakes which were committed involuntarily over this point. The English was revised by a native speaker, so we have made a thorough review of the writing of the work, thereby improving the language of the manuscript (English). On the other hand, the words written in British and American English were changed according to the suggestions made by the reviewer. In the improved version of the writing, you can verify the changes made to the research work. 

2        Although the theory and the simulation results are compatible, the authors must present experimental validation of the interrogation method in order to demonstrate its feasibility. There are several issues that must be considered in practical measurements, such as the algorithm processing time, the robustness to environmental noise, the imperfections in the FBG and FPI dimensions, etc.

Answer:

The comment is quite meaningful. Allow me to expose that unfortunately, in this work, the experiments of the proposed methodology were not considered. This, due to the fact that at this moment we do not have a way to perform such experiments as suggested by the reviewer. Currently, our research group is doing everything possible to install an experimental system and thus be able to make the necessary measurements to corroborate our results. It should be noted that the approach of this research work is to propose a theoretical and computational methodology, which will allow us to explain the capacity of the quasi-distributed fiber optic sensor based on the FDM / WDM multiplexing and Fabry-Pérot interferometric sensors. The theory and simulations consider important factors of the optical sensing system such as optical instrumentation, parameters of local Fabry-Pérot sensors, detection technique, signal demodulation algorithm, multiplexing techniques and noise in the system. Not all possible factors are considered because the problem may end up with no solution.

Reviewer 3 Report

Review article:

Signal analysis, signal demodulation and numerical simulation of a quasi-distributed optical fiber sensor based on FDM/WDM techniques and Fabry-Pérot interferometers.

The authors present a work about a Signal analysis, signal demodulation and numerical simulation of a quasi-distributed optical fiber sensor based on FDM/WDM techniques and Fabry-Pérot interferometers.

This is a well-written and structured manuscript. In this manuscript there are the potentialities to publications in Sensors in this form.

Minor revision.

Between rows 60-65 the numbers 2 wavelength, 4 frequency channels, 3 wavelength, 9 Fabry-Pérot can be changed in two wavelength, four frequency channels, three wavelength, nine Fabry-Pérot

Author Response

Response to Reviewer 3 comments:

Between rows 60-65 the numbers 2 wavelength, 4 frequency channels, 3 wavelength, 9 Fabry-Pérot can be changed in two wavelength, four frequency channels, three wavelength, nine Fabry-Pérot

Answer:

The corrections suggested by the reviewer were done.

Correction:

The numbers were changed by words between rows 60-65, page 2 and paragraph 2.

Round  2

Reviewer 2 Report

The authors managed to address most of the reviewers' queries in this new version, but the experimental validation of the methodology is still missing.
However, according to the response, it is not possible to conduct such measurements in practical time.
The manuscript can be published in the present form because the journal accepts pure theoretical results, but I strongly advise the group to validate the proposed technique with experimental data in further papers in order to make it more convincing.